# Development of Stem Cell-Derived Immune Cells for Off-the-Shelf Cancer Immunotherapies

**DOI:** 10.3390/cells10123497

**Published:** 2021-12-10

**Authors:** Yan-Ruide Li, Zachary Spencer Dunn, Yang Zhou, Derek Lee, Lili Yang

**Affiliations:** 1Department of Microbiology, Immunology & Molecular Genetics, University of California, Los Angeles, CA 90095, USA; charlie.li@ucla.edu (Y.-R.L.); zzydcat@g.ucla.edu (Y.Z.); ylee932@ucla.edu (D.L.); 2Mork Family Department of Chemical Engineering and Materials Science, University of Southern California, Los Angeles, CA 90089, USA; zacharsd@usc.edu; 3Eli and Edythe Broad Center of Regenerative Medicine and Stem Cell Research, University of California, Los Angeles, CA 90095, USA; 4Jonsson Comprehensive Cancer Center, David Geffen School of Medicine, University of California, Los Angeles, CA 90095, USA; 5Molecular Biology Institute, University of California, Los Angeles, CA 90095, USA

**Keywords:** stem cell engineering, allogeneic cancer therapy, off-the-shelf cell therapy, chimeric antigen receptor (CAR), T cell receptor (TCR), graft-versus-host disease (GvHD)

## Abstract

Cell-based cancer immunotherapy has revolutionized the treatment of hematological malignancies. Specifically, autologous chimeric antigen receptor-engineered T (CAR-T) cell therapies have received approvals for treating leukemias, lymphomas, and multiple myeloma following unprecedented clinical response rates. A critical barrier to the widespread usage of current CAR-T cell products is their autologous nature, which renders these cellular products patient-selective, costly, and challenging to manufacture. Allogeneic cell products can be scalable and readily administrable but face critical concerns of graft-versus-host disease (GvHD), a life-threatening adverse event in which therapeutic cells attack host tissues, and allorejection, in which host immune cells eliminate therapeutic cells, thereby limiting their antitumor efficacy. In this review, we discuss recent advances in developing stem cell-engineered allogeneic cell therapies that aim to overcome the limitations of current autologous and allogeneic cell therapies, with a special focus on stem cell-engineered conventional αβ T cells, unconventional T (iNKT, MAIT, and γδ T) cells, and natural killer (NK) cells.

## 1. Introduction

After decades of fervent research, tumor-targeting adoptive T cell therapy has entered mainstream oncology [1]. In the 1980s, Rosenberg and others conducted numerous trials testing autologous tumor infiltrating lymphocyte (TIL) therapy and witnessed notable although rare clinical responses in certain chemotherapy refractory cancers [2,3,4]. Advances in molecular engineering ushered in a new era of adoptive therapy in which tailor-made T cells are genetically modified with the machinery to both target and kill cancer cells [5]. Chimeric antigen receptors, or CARs, link the single chain variable fragment of an antibody to T cell intracellular activation and stimulatory domains, allowing T cells to recognize cancer cells independently of major histocompatibility complex (MHC) restriction and perform cytotoxic functions [6]. CAR-T cells have transformed the treatment of blood cancers, with CD19-targeting CAR-T cells approved for treating B cell acute lymphoblastic leukemia and large diffuse B cell lymphoma in 2017 and a BCMA-targeting CAR-T cell therapy approved in 2021 for the treatment of multiple myeloma [7]. The current CAR-T cell therapies are autologous and, while landmark achievements for cell therapy, limited in their accessibility. T cell extraction, genetic manipulation, expansion, and reinfusion for each individual result in patient-to-patient variability, patient selectivity (time to manufacture, access to facilities), and exorbitant costs [8,9]. Furthermore, patient pretreatment and status, and the rapid expansion of immune cells, can result in highly differentiated and low-quality final products that limit therapeutic efficacy [10,11].

Allogeneic cell therapies, shown in Figure 1, hold promise as accessible, readily administrable products selected for desirable clinical attributes but face critical safety and efficacy concerns, namely, graft-versus-host disease and host-versus-graft rejection, respectively [12]. Much of our knowledge of graft-versus-host disease comes from long-standing experience with allogeneic hematopoietic stem cell transplants (Allo-HSCT) for the treatment of hematological malignancies and other blood disorders [13]. Allo-HSCT is the first clinically validated cancer immunotherapy and remains the only curative option for several blood cancers. Within Allo-HSCT grafts, donor T cells exert potent graft-versus-tumor effects, but these same effector cells can recognize major and minor HLA complex mismatches and attack healthy host tissue. Graft-versus-host disease (GvHD) occurs in 30–70% of Allo-HSCT patients, which can limit the therapeutic benefit of this treatment [14,15,16,17,18]. Although depleting these T cells from allografts reduces the risk of GvHD, it leads to increased rates of tumor relapse and graft failure and is not performed in clinical care [19,20]. Typical GvHD prophylaxis and treatment consists of standard immunosuppressive medication such as calcineurin inhibitors (ciclosporin or tacrolimus) and/or methotrexate, anti-T-lymphocyte globulin (ATG), and post-transplant cyclophosphamide, as well as steroids and several candidates for steroid-refractory GvHD, including ibrutinib (which is approved by the FDA), alemtuzumab, JAK inhibitors, rituximab, mammalian target of rapamycin (mTOR) inhibitors, and others [15]. The evolution of immunosuppressive medications has greatly improved the management of GvHD, and Allo-HSCT grafts can establish host-versus-graft (HvG) tolerance and achieve durable engraftment of donor cells [17]. For non-hematological compartment reconstituting therapies, such as allogeneic CAR T cell therapies, HvG responses, while not life-threatening, can weaken the cell therapy before it fully executes antitumor functions [21]. Once again, scientific advancements are opening the doors for new and improved cells therapies, including the creation of allogeneic mature immune cell adoptive treatments that can avoid GvHD as well as HvG responses. CRISPR knockout can be used to remove endogenous TCRs, alleviating GvH concerns, as well as HLA Class I and II molecules (B2M and CIITA knockouts), rendering the cells resistance to immunorejection by host T cells [22]. HLA-E and other NK cell inhibitory receptors can be incorporated into cells to mitigate NK cell-mediated elimination [23,24] Importantly, conventional αβ T cell-based universal CD19-CAR-engineered T cells (UCART19) using CRISPR KO of the TCRα chain have proven to be safe in the clinic and showed notable antitumor efficacy with an objective response rate (ORR) of 67% [25]. Although the ORR and duration of responses were smaller than those of autologous CAR19 T cells of comparable design, likely due to shorter persistence of the universal CAR-T cells, all 21 patients enrolled received their scheduled UCART19 treatment [26].

Parallel to the rapid expansion of allogeneic conventional T cell therapy research and investment is the development of adoptive transfer strategies using other cell populations (Figure 2) [12,27]. In contrast to conventional αβ T cells, natural killer (NK) cells and innate-like T cells, such as gamma delta (γδ) T, invariant natural killer T (iNKT), and mucosal associated invariant T (MAIT) cells, do not bind peptide-MHC complexes and therefore pose little risk of GvHD (Table 1). NK cells express germline-derived activating and inhibitory receptors that allow recognition of missing-self, which makes NK cells instrumental in cancer immunosurveillance, as tumor cells often alter MHC expression to evade T cell immunity [28]. Several NK cell-targeting antibodies and adoptive therapies seek to harness the inherent cancer-killing ability of NK cells and incorporate additional activation signals [29]. Following phase 1/2 studies establishing the safety and feasibility of allogeneic NK cells for cancer treatment [30], CAR-engineered cord-blood (CB)-derived NK (CB-NK) cells have entered the clinic to increase therapeutic efficacy. This past year, Liu et. al. reported a 73% response rate and excellent safety profile of CAR-transduced CB-NK cells in CD19-positive lymphoid tumors [31]. γδ T, iNKT, and MAIT cells recognize phosphoantigens, glycolipids, and microbial vitamin B_2_ (riboflavin) biosynthesis bioproducts, respectively, allowing the targeting of numerous cancer cells through TCR-dependent mechanisms [32]. These unconventional T cells also express innate killer receptors, such as NKG2D, and can rapidly release cytokines upon stimulation [33]. Similar to standard T cells, innate-like T cells are amenable to genetic engineering and are compatible with CAR expression. Gamma delta and iNKT cell therapies have progressed to clinical trials [32], with Heczey et al. recently reporting signs of clinical activity of autologous GD2-targeting CAR iNKT cells in pediatric neuroblastoma patients [34] and Xu showing that, in 132 late-stage cancer patients, allogeneic Vγ9Vδ2 adoptive T-cell immunotherapy was safe and prolonged the survival of patients treated with multiple doses [35].

Autologous iNKT cell therapies face the same challenges as autologous conventional αβ T cell therapy, with an additional hurdle mounted by the rarity of iNKT cells, which account for about less than 1% of peripheral blood mononuclear cells [62,63,64,65]. γδ T cells, although to a lesser extent, are also scarce in the periphery (5% of PBMCs) [66]. Strategies that use mature immune cells as the product starting material also struggle to create homogenous and fecund cell products as a result of gene transduction and knockout inefficiencies, variability in the initial cell composition, and exhaustive expansion procedures. Despite these hurdles, allogeneic cell therapies created from fully differentiated conventional αβ T, γδ T, iNKT, and NK cells are being actively pursued in early phase clinical trials, where thus far the therapies have displayed encouraging safety profiles and signs of efficacy.

Stem cell engineering has emerged as a novel solution to address the limitations faced by current autologous and allogeneic cell therapies. Stem cells can undergo multiple gene edits and expand clonally to produce pure, high quality effector cells. Recent progress in stem cell culture and differentiation has resulted in the burgeoning development of stem cell-derived adoptive cellular candidates. In this review, we highlight hematopoietic and pluripotent stem cell engineering methods and their ability to produce effective and safe products.

## 2. Stem Cell Resources and Culture Systems

Two major categories of stem cell resources are used for developing allogeneic therapeutic cells: hematopoietic stem cells (HSCs) and pluripotent stem cells (PSC). Multipotent HSCs can be collected from umbilical cord blood (UCB), donor bone marrow, and granulocyte colony stimulating factor-mobilized peripheral blood [67,68]. Established PSC lines, including embryonic stem cell (ESC) and induced pluripotent stem cell (iPSC) lines, are widely utilized to differentiate and generate to hematopoietic stem/progenitor cells and mature immune cells. In addition, peripheral blood mononuclear cell (PBMC)-derived immune cells including T, NK, iNKT, and MAIT cells could be reprogramed to pluripotency and then re-differentiated into functional immune cells [38,55,69,70].

Various culture systems were developed to support stem cell differentiation, including the humanized mouse models (e.g., bone marrow-liver-thymus, BLT mouse model), in vitro feeder-dependent culture systems (e.g., OP9-DL and artificial thymic organoid, ATO), and in vitro feeder-free culture systems.

The in vitro OP9-DL system relies on a genetically engineered murine bone marrow stromal cell line OP9, which overexpresses the Notch ligands Delta-like ligand 1 (DLL-1) or 4 (DLL-4) [71,72,73]. This culture system supports the efficient generation of human HSC-derived T and NK cells [74,75]. To differentiate into T cells, ESCs or iPSCs are first co-cultured with C3H10T1/2 stromal cells for efficient hematopoietic stem/progenitor cells (HSPCs) generation [76,77] and then co-cultured with OP9-DL1 stromal cells to ignite Notch signaling for T-lineage commitment [78,79]. The cells are finally mixed with PBMCs for stimulating mature T cell proliferation [70]. To differentiate into NK cells, ESCs or iPSCs are cultured in a stromal cell-based or a stromal-free system supplemented with stem cell factor (SCF), vascular endothelial growth factor (VEGF), and bone morphogenetic protein 4 (BMP4) to induce hematopoietic differentiation and then are switched to cultures containing IL-3, IL-15, IL-7, SCF, and FLT3L to stimulate NK cell differentiation, followed by propagation with a stimulatory cell line expressing membrane-bound IL-15 (mbIL-15) or mbIL-21 [56,57,80,81,82].

The in vitro ATO culture system was developed by Dr. Crooks team at UCLA and has been used for generating human T cells from HSCs or PSCs [40,41]. ATO supports T cell differentiation by mimicking natural human T commitment [40,41] and ATO-derived mature T cells exhibit a highly diverse TCR repertoire, an antigen-naïve phenotype, and a vigorous response to antigen stimulation. Genetically engineered stem cells could also be cultured in ATO system and differentiated into TCR-engineered, antigen-specific T cells [40,41].

The “off-the-shelf” in vitro generation of human T cells has been an important approach for studying T cell development and applying this to T cell-based immunotherapy. However, due to mouse origins, OP9-DL and ATO culture systems have not been used for clinical studies. Two strategies have been developed to circumvent the potential issue: (1) design feeder cells of human origin that can support T cell development similar to OP9-DL cells; (2) create a feeder-free culture system where all the molecular necessities supporting T cell development are supplied with defined media, cytokines, and reagents with minimal animal or human origins. Remarkably, a Notch signaling-dependent ex vivo differentiation/expansion system using feeder-free/serum-free Stemspan media has been studied [83]. This system supports the development of human hematopoietic stem/progenitor cell-derived immune cells and the engraftment of these immune cells into humanized mice [83]. However, clinical trials showed that the CD34^+^ stem/progenitor cells expanded ex vivo in the presence of Notch ligand led to a rapid myeloid reconstitution post adoptive transplant, rather than T cell lineage [83]. Further improvements are necessary to achieve T cell reconstitution and expansion.

The BLT mouse (human bone marrow-liver-thymus engrafted NOD/SCID^γc−/−^ mouse) model was created by co-transplanting human CD34^+^ HSCs, liver, and fetal thymus into humanized immunodeficient mice. BLT provides a humanized mouse carrier supporting human immune system establishment and human immune cell generation [84,85]. TCR-engineered antigen-specific T cells can be generated by transducing HSCs using lentivirus or retrovirus and adoptively transferring these HSCs to BLT mice [64,86,87,88]. The BLT model can also be utilized as a valuable tool to study the biology and translational potential of human HSC-derived T cells. However, these generated T cells are educated in the transplanted human thymus and they do not develop tolerance to the BLT mouse host; therefore, these self-reactive T cells eventually cause GvHD and host fatality [89]. To overcome this issue, sub-lethally irradiated neonatal mice were used as a new BLT model, where the transplanted human T cells were educated in the host thymus, leading to a restricted TCR repertoire and improved safety profile compared with the previous BLT model [64,89]. This new model likely contributes to the host-tolerant mature human T cells and allows long-term studies of these humanized animals. In addition, using NSG hosts engineered to express homozygous human HLA class I heavy chain and light chain can allow the generation of an HLA-restricted T-cell repertoire [90].

Several studies have reported the approaches reprogramming PBMCs to pluripotent iPSCs. Human T-lineage cells, such as antigen-specific cytotoxic T cells, invariant natural killer T (iNKT) cells, or mucosal associated invariant T (MAIT) cells, are transduced with defective Sendai virus vectors encoding four reprogramming factors (OCT3/4, SOX2, KLF4, and c-MYC) [91] and SV40 T antigen to be reprogrammed into pluripotency [43,46,49,50]. Non-T cell-derived PBC-iPSCs are used for enhanced NK cell commitment [92].

## 3. Allogeneic Stem Cell-Engineered T Cell-Based Therapy

Allogeneic T cell therapy, especially allogeneic CAR-T therapy, has attracted much attention because of the great advantage of wide and prompt usage for patients. The two main hurdles of widely using allogeneic T cell-based therapy are the risks of inducing GvHD and being rejected by the host [9]. To overcome these issues, multiple genes including *TRAC*, *B2M*, and *PDCD1* were depleted in CAR-T cells to enhance their antitumor activity and decrease risk of GvHD and host allorejection [23,93]. Various strategies have been applied to improve the manufacture, cancer-treating potential, and safety of allogeneic T cell products, including applying base editor technology to mediate highly efficient multiplex gene disruption with minimal double-strand break induction [94] and targeting the insertion of a CAR Transgene directly into the native TCR locus using an engineered homing endonuclease and an adeno-associated virus (AAV) donor template [37]. Notably, one allogeneic cell product, UCART19, was recently tested in phase I clinical trials to treat CD19^+^ B cell malignancies [25,95]. The UCART19-based therapy was developed to ablate the endogenous αβ TCR of CAR-T cells to diminish GvHD, lymphodeplete to reduce host cell-mediated allorejection, and disrupt the CD52 of CAR-T cells to grant the cells resistance to lymphodepleting drugs [25,95].

Alternative methods to generate allogeneic CAR-T cells without expression of the endogenous TCR are gene engineering of stem cells. Because of allelic exclusion, T cells generated from TCR-transgenic hematopoietic progenitor cells do not rearrange endogenous TCR loci and express only the transgenic TCR, leading to a reduced risk of inducing GvHD [96,97]. The development of stem cell-derived allogeneic CAR-T therapy involves the transduction of stem cells to express a tumor-specific TCR (e.g., NY-ESO-1 and MART1-specific TCRs) or CARs and subsequent differentiation of the stem cells to T cells in stem cell culture system (e.g., OP9-DL, ATO, and feeder-free systems) [38,39,40,41,96]. These stem cell-derived T cells display specific cytokine production upon activation, potent antitumor capacity, and limited occurrence of GvHD [38,39,40,41,96].

Several studies have investigated the production of iPSCs from antigen-specific T cells from patients [43,46,70]. Researchers reprogrammed antigen-specific CD8^+^ cytotoxic T cells to pluripotency and then re-differentiated these T cell-derived iPSCs into mature CD8^+^ T cells. These “rejuvenated” cytotoxic T cells demonstrated specific reactivity upon the same antigen stimulation and displayed TCR gene-rearrangement patterns identical to those of the patient’s original CD8^+^ T cells [43,46,70]. The unlimited resources of T cell-derived iPSCs illustrate a strategy generating functional antigen-specific CD8^+^ T cells that might be applicable in cancer immunotherapy.

Engineering stem cells also provides an efficient approach to generate off-the-shelf therapeutic cells without the rejection due to recognition by host T cells or NK cells. Knock-out or knock-down of MHC molecules has been explored to avoid host T cell-mediated allorejection [98]. However, the lacking MHC expression on therapeutic cells may induce the target and elimination by host NK cell [98]. Expression of the ligands to NK inhibitory receptors, such as HLA-E or HLA-G, can further increase the resistance of engineered cells to host NK cell-mediated allorejection [24,99,100,101,102]. Overall, the advantages of stem cell-derived CAR-T cells including large-grade manufacturing and relative ease of genomic modification, provide the potential to generate ready-to-use cell banks as standardized “off-the-shelf” immunotherapies to treat blood cancers and solid tumors using different CAR constructs.

## 4. Allogeneic Stem Cell-Engineered Unconventional T Cell-Based Therapy

While conventional αβ T cells have been utilized for generating allogeneic cell products by ablating their endogenous TCR expression, exploring third-party off-the-shelf strategies that do not require genome editing for safe administration is intensively appealing. Unconventional T cells, such as lipid-restricted invariant natural killer T (iNKT) cells, MR1-restrict mucosal associated invariant T (MAIT) cells, and gamma delta T (γδ T) cells harbor unique features that could potentially qualify them as universal donor cells for cancer immunotherapy.

### 4.1. Allogeneic iNKT Cell-Based Therapy

iNKT cells are a distinctive T cell subpopulation expressing semi-invariant TCRs that recognize lipid antigens in the context of monomorphic antigen-presenting molecule CD1d [12,103,104]. The restricted TCR is comprised of a canonical invariant TCRα chain (Vα14-Jα18 in mice; Vα24-Jα18 in human) paired with a semi-variant TCRβ chain (mostly Vβ8.2 in mice; mostly Vβ11 in human). Upon TCR engagement, iNKT cells can upregulate killing receptors (e.g., FasL, TRAIL) and rapidly secret cytotoxic molecules (perforin and granzymes) and high levels of cytokines (e.g., IFN-γ, TNF-α, IL-2, IL-4, IL-17), leading to the activation of both innate and adaptive immune cells. Thus, they can rapidly attack tumor cells through multiple mechanisms [105,106] and strongly modulate the tumor microenvironment [88,107,108]. Their capacity to mount strong anti-tumor responses without inducing GvHD makes them an attractive candidate for cancer immunotherapy [109,110,111,112]. The widespread application of iNKT cell-based cancer therapy is severely hindered by the extremely low frequency of iNKT cells in the peripheral blood. Although clinical trials have focused on administrating a-GalCer/a-Galcer-pulsed dendritic cells (DC) to boost endogenous iNKT cell numbers or adoptively transferring ex vivo expanded iNKT cells to restore iNKT cell functions, the responses were not as encouraging as expected [113,114,115]. The current GMP-compatible ex vivo expansion protocols are now being used to expand autologous iNKT cells with CAR engineering, which produced very promising results in pre-clinical studies on treating neuroblastoma and B cell lymphoma [116,117,118,119]. The safety and long-term persistence are still under clinical evaluation. Since stem cells possess unique properties for creating allogeneic cell therapies, using stem cell-derived iNKT cells or CAR-iNKT cells is an active area research. The Yang, Kaneko, and Taniguchi groups have reported the successful production of human iNKT cells by the genetic engineering of HSCs or differentiation iPSCs, and the iNKT cells were responsive to a-Galcer stimulation and executed potent anti-tumor capability toward leukemia, multiple myeloma, and solid tumors [9,49,64,65,88,120]. These pre-clinical results provide promising support for the development of iNKT cells as allogeneic third-party universal donors to change the paradigm of cancer immunotherapy.

### 4.2. Allogeneic MAIT Cell-Based Therapy

Mucosal-associated invariant T (MAIT) cells are another innate T lymphocyte population. They express a semi-invariant TCR, consisting of an invariant TCRVα chain paired with a limited number of Vβ chains [121,122]. MAIT TCRs recognize riboflavin metabolite-based antigens and folate derivatives presented by an evolutionary conserved and monomorphic protein MR1 [9,12]. MAIT cells constitute 5% of the total T cell population in humans and exhibit tissue-specific distribution. TCR engagement of MAIT cells lead to the secretion of perforin, granzyme B, and other TH1 and TH17 type of cytokines [122]. Studies have showed that MAIT cells are part of tumor-infiltrating lymphocytes in cancer patients [123,124,125,126,127], although it remains controversial whether tumor-infiltrating MAIT cells are pro- or anti-tumorigenic [128]. Further investigations are necessary to elucidate the role of MAIT cells in cancer progression. Since MAIT cells are not MHC-restricted, they can be another candidate for developing allogeneic cell therapy. Compared to other unconventional T cells, MAIT cells are still not well studied.

iNKT, MAIT, and γδ T cells have demonstrated strong antitumor ability independent of MHC-restriction. Furthermore, the unique feature that they are not expected to induce GvHD risks provides them with great promise for developing allogeneic cell therapy to treat cancer. It has been shown that genetic engineering of HSCs or iPSC-reprogramming can successfully generate allogeneic iNKT cells, γδ T, and MAIT cells [49,50,64,120]. While production yield is still a critical hurdle that limits the manufacturing of these unconventional T cell products, there are novel cell culture systems (e.g., ATO, feeder-free culture system) that may allow large-scale production of ‘off-the-shelf’ cell products [40,41,129,130]. These approaches will provide platforms for studying the potential of iNKT, MAIT, and γδ T cell-based allogenic therapies and set up a foundation for future off-the-shelf cancer immunotherapy.

### 4.3. Allogeneic γδ T Cell-Based Therapy

Another promise candidate for developing off-the-shelf cell therapy is butyrophilin (BTN)-restricted Vγ9Vδ2T cells [122,131]. This unique subpopulation cells represents 0.5–5% of all T cells and 50–90% of γδ T cells [132,133]. Vγ9Vδ2T cells express an invariant TCR that responds to phosphoantigens (pAgs) or phosphorylated isoprenoid metabolites that are derived from the mevalonate pathway. These pAgs are widely expressed on transformed or infected cells that have dysregulated metabolism [134]. pAgs bind to the intracellular domain of BTN3A and induces the activation of Vγ9Vδ2T cells. The activated Vγ9Vδ2T cells display similar effector functions as conventional αβ T cells that secrete perforins and granzymes and produce pro-inflammatory cytokines to directly kill tumor cells and modulate immune responses [133,134]. The activated Vγ9Vδ2T cells themselves can also differentiate into professional APCs that can phagocytose cells and cross-present antigens, leading to the activation of conventional T cells [135,136]. The current approach to expand PBMC Vγ9Vδ2T cells in vitro is the use of a synthetic aminobisphosphonate drug, Zoledronate [137,138,139]. Zoledronate stimulation can generate clinically reasonable numbers of functional Vγ9Vδ2T cells that are able to migrate to tumor sites and perform tumor cell killing. The effects of allogeneic γδ T cell therapy on blood cancers and solid tumors are still under investigation (NCT03533816, NCT03790072). If the clinical safety of allogenic Vγ9Vδ2T cells is validated, similar to iNKT cells, engineering stem cells with γδ TCR to produce pure and clonal cells can further facilitate γδ T cell-based therapy.

## 5. Allogeneic Stem Cell-Engineered NK Cell-Based Therapy

In recent years, CAR-engineered NK cells have gained enormous attention because of their unique properties fitting for cancer immunotherapy. NK cells are innate immune cells showing strong cytotoxicity against physiologically stressed cells such as tumor cells and virus-infected cells through multiple mechanisms of action. Their recognition of the target cell is independent of MHC expression [140]. NK cell activation and effector functions rely on the signals derived from both activating and inhibitory receptors. Activating signals include cytokine-binding receptors, integrins, killing-receptors (e.g., CD16, NKp30, NKp40, and NKp44) [140]. Inhibitory signals mainly come from receptors recognizing MHC-I, as well as some MHC-I non-related receptors [141]. In addition, MHC-I inhibitory receptors can be divided into three categories based on structure and function: killer lectin-like receptors (KLRs), killer cell immunoglobulin-like receptors (KIRs), and leukocyte immunoglobulin-like receptors (LILRs) [141].

NK cells, even when genetically engineered CAR molecules, retain the capacity to target tumor cells through their intrinsic activating receptors, thereby granting them with an additional mechanism of anti-tumor reactivity independent of CAR-mediated killing. Notably, NK cells do not rely on the TCR for cytotoxic killing, and this feature endows NK cells with a more favorable safety profile compared to T cell contenders, which, in the allogeneic setting, need to be further modified to diminish GvHD [142].

### 5.1. CAR-NK Cells Derived from Umbilical Cord Blood (UCB)

CAR-NK cells can be generated from different sources. UCB is a readily available source for allogeneic NK cell production [143]. Although the starting cell numbers are low, they can be easily expanded to large, highly functional products due to their inherent capacity of high proliferation. The first large-scale clinical trial of CD19 CAR-engineered NK therapy was performed on 11 chronic lymphocytic leukemia and non-Hodgkin’s lymphoma patients in MD Anderson cancer center [31,142]. All patients were treated with lymphodepleting chemotherapy before CAR-NK infusion. Patients received UCB-derived CD19 CAR-NK cells containing a suicide gene switch and an immune enhance gene *IL-15*. Seven out of 11 patients responded well and experienced sustained complete remission up to 13.8 months. Notably, the infused CAR-NK cells were able to persist in the patient blood over one year. This clinical study exhibited the administration of UCB-derived CD19 CAR-engineered NK therapy in B cell lineage malignancies to be efficacious and safe [31].

### 5.2. CAR-NK Cells Derived from Other PSCs

CAR-NK cells can also be generated from PSCs including ESCs and iPSCs [144]. In 2005, Woll et al. generated human ESC-derived NK cells using a two-stage culture system. These cells resembled endogenous NK cells, targeted tumor cells using multiple mechanisms including direct cell-mediated cytotoxicity and antibody-dependent cellular cytotoxicity (ADCC), and displayed powerful antitumor capacity in vivo [57,82]. The same group also developed a novel platform to produce NK cells from iPSCs [80,145,146]. The NK cells were developed from a clonal master iPSC line cell bank, making it feasible to mass generate iPSC-NK cells, which are relatively homogenous, quality controlled, and able to be cryopreserved for long-term storage. iPSCs were first genetically modified to express or knockout the genes of interest, and then they were made into aggregates by centrifugation to form embryoid bodies [80,145,146]. After differentiating into CD34^+^CD45^+^ hematopoietic progenitor cells, they were further differentiated into mature NK cells using a specific cytokine cocktail. The generated iPSC-derived NK cells displayed common NK cell markers, including NKG2D, NKp44, NKp46, KIRs, CD16, and TRAIL, and these cells were cytotoxic against hematological and solid tumor cells in vitro and in vivo [80]. Next, iPSC-derived NK cells were stimulated and expanded using cytokines and K562-based artificial APCs with membrane-bound IL-21 to achieve high yield for clinical and translational applications. These iPSC-derived NK cells could be further engineered with either conventional T cell CARs or NK cells CARs containing the transmembrane domain of NKG2D to enhance their tumor targeting abilities [56]. A high-affinity noncleavable CD16a (hnCD16) was engineered on iPSC-derived NK cells to improve their ADCC properties, and the hnCD16-engineered NK cells combined with mAbs showed highly effective killing of hematologic malignancies and solid tumors [59].

Currently, Fate Therapeutics Company is conducting clinical trials based on the iPSC-derived NK products [146]. For example, FT596 is an investigational, universal, off-the-shelf iPSC-derived NK cell product engineered with hnCD16 and CD19 CAR (denoted as CAR.19-NKG2D-2B4-CD3ζ-IL15RF-hnCD16) [140]. The clinical trial is studying the efficacy of FT596 monotherapy and a combination of FT596 with a CD20 monoclonal antibody in the treatment of chronic lymphocytic leukemia and B cell lymphoma. The Phase I interim result showed the treatments were well-tolerated, with no dose-limiting toxicities, and 10 out of 14 patients achieved the objected response [140].

## 6. Outlook

Autologous cell therapy has transformed the treatment of hematological malignancies. Patients with relapsed and refractory B cell cancers experience response rates of up to 90% with CAR-T cell treatment, and durable clinical benefit occurs in about 30–40% of patients [147]. Tumor-infiltrating lymphocyte (TIL) therapy continues to show clinical promise, with durable responses in some patients with refractory melanoma and cervical squamous cell carcinoma [148,149], and recombinant TCR-transduced T cells have shown encouraging clinical activity in multiple myeloma and melanoma [150]. These autologous therapies are by nature one of one, which hinders their manufacturability, accessibility, and affordability. Current CAR-T cell therapies are priced over $300,000 per treatment, not including additional costs associated with adverse events, and typically require over two weeks for production and administration [151]. Patients with rapidly progressing disease may not qualify for CAR-T cell therapy, and patient-derived cell starting material results in highly variable final products [152]. Allogeneic cell sources enable “off-the-shelf” cell therapies that can be produced at scale and administered on demand but face severe challenges of their own [153]. Graft-versus-host responses of conventional αβ T cells require efficient gene-editing of T cells or the use of non-alloreactive cell populations. Allogeneic cells also confront the host immune system, which can limit the persistence and efficacy of donor-derived cells. Various cell populations have been studied to achieve the holy grail of allogeneic cell therapy: maximizing the cancer-fighting ability of allogeneic cells while minimizing GvHD and allorejection. In this review, we highlighted the potential of stem cell-engineered immune cell populations other than conventional αβ T cells, specifically NK, γδ T, iNKT, and MAIT cells, to achieve this aim. Of note, mesenchymal stem cells (MSCs) have also been investigated for cancer treatment [154,155]. MSCs can be used as cell carriers for targeted cancer therapy given their immune evasive and migratory properties.

The intrinsic genomic instability of cancer cells coupled with the Darwinian process of immunoediting precipitates cancer cells that can avoid immune destruction [156,157]. Antigen negative relapse has been documented in CAR-T cell therapy [158], as well as the loss of an immunogenic epitope following TIL therapy [159]. We propose that multiple tumor-killing mechanisms are vital for adoptively transferred cells to contend with a cancer’s plasticity and heterogeneity. NK, γδ T, iNKT, and MAIT cells all possess intrinsic cancer-killing ability. Using these cellular populations as carriers for CARs thus enables the killing of CAR-antigen positive and negative tumor cells.

The ex vivo activation, genetic manipulation, and expansion of patient or healthy donor lymphocytes lead to the differentiation of effector cells to achieve necessary cell numbers for dosing. This can result in cell products with limited self-renewal potential and diminished persistence upon infusion [10,160]. Telomere, differentiation, and CDKN2a mRNA analysis revealed that 15 days of T cell expansion aged cells the equivalent of 30 years [161]. Initiating genetic engineering and immune cell development at the stem cell level gives researchers control over the differentiation status of the final cell product while maintaining production of sufficient cell numbers. The massive expansion of stem cells can make multi-, high-dose strategies possible for all patients.

The number of genetic alterations that can be successfully applied to stem cells is continuing to grow. Wang et al. recently reported the generation of hypoimmunogenic T cells from genetically engineered allogeneic human iPSCs, in which iPSCs lacking MHC Class I, MHC Class II, and NK cell-ligand poliovirus receptor CD155 were transduced to express single-chain MHC Class I antigen E [162]. Following iPSC to T cell differentiation, the resulting T cells were resistant to T, B, and NK cell alloreactivity and, when further manipulated to express CAR, controlled preclinical tumor growth. The next steps are to incorporate modifications that allow the adoptively transferred cells to persistent autonomously, maintain proliferative potential, outmaneuver the immunosuppressive tumor microenvironment, infiltrate tumor beds, and stimulate endogenous antitumor immunity. Each of these goals has been addressed extensively in preclinical T and NK cell studies [163,164,165,166], such as through the exogenous expression of IL-15, immune checkpoint inhibitors, chemokine receptors, or immunomodulatory proteins, but are usually targeted individually or in pairs due to the limited genetic pliability of mature immune cells. Stem cell engineering opens the door for increasingly complex designer cell products, and future research will need to reveal if the accumulated changes hinder immune cell antitumor efficacy. By pursuing allogeneic therapies using stem cell-derived NK, gamma delta T, iNKT, and MAIT cells, we can take advantage of their natural tumor-targeting abilities and superior safety profiles to create ideal candidates for off-the-shelf cancer cell therapies.

## Figures and Tables

**Figure 1 cells-10-03497-f001:**
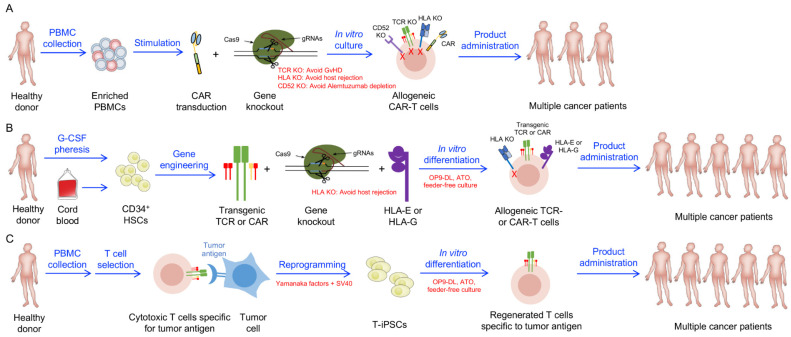
Current allogeneic T cell-based cancer immunotherapies. (**A**) PBMCs are collected from healthy donors via leukapheresis and then are genetically engineered. CARs are transduced into target cells (e.g., T, NK, or iNKT cells) via Lenti or Retrovirus. CRISPR-Cas9-mediated gene editing is used to knock out genes encoding TCR, HLAs, and CD52 to lessen the GvHD risk, HvG risk, and anti-CD52 monoclonal antibody alemtuzumab-induced cell depletion, respectively. (**B**) Human CD34^+^ HSCs are collected from either cord blood or from G-CSF-mobilized human peripheral blood. These HSCs are transduced with transgenic TCRs or CARs and other molecules (e.g., HLA-E and HLA-G) and then engineered with a CRISPR-Cas9/gRNAs complex to knockout HLAs. The gene-engineered HSCs are put into “off-the-shelf” in vitro culture systems including OP9-DL, ATO, or feeder-free culture systems to differentiate into mature immune cells. Of note, gene engineering and editing steps could be performed on stem cells or differentiated mature immune cells. Performing gene-engineering and/or gene-editing on stem cells could save on the use of gene-engineering/editing materials such as lentivectors and CRISPR-Cas9/gRNAs and also enable the maximal gene engineering/editing efficiency, which can be carried on into the final cell products. (**C**) Clonally expanded tumor antigen-specific T cells are reprogrammed to pluripotency. These T cell-derived iPSCs are then re-differentiated into mature T cells in vitro. These “rejuvenated” T cells may have potentials in the field of adoptive and allogeneic immunotherapy. Abbreviations: PBMC, peripheral blood mononuclear cells; CAR, chimeric antigen receptor; KO, knockout; G-CSF, granulocyte-colony stimulating factor; iPSC, induced pluripotent stem cells; SV40, simian vacuolating virus 40.

**Figure 2 cells-10-03497-f002:**
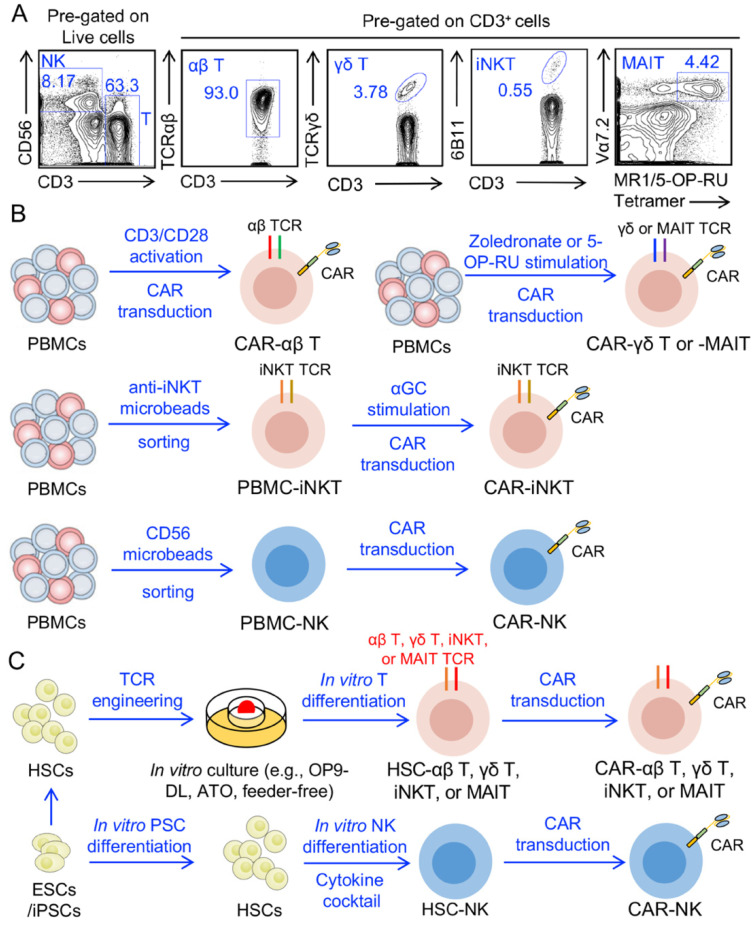
Engineering stem cells to generate allogeneic CAR-expressing αβ T, γδ T, iNKT, MAIT, and NK cells. (**A**) FACS plots showing the analysis of cells from healthy donor PBMCs. Conventional αβ T, γδ T, iNKT, MAIT, and NK cells were analyzed. (**B**) Healthy donor PBMCs are used to generate the CAR-engineered conventional αβ T, γδ T, iNKT, MAIT, and NK cells. To generate conventional αβ T cells, PBMCs are stimulated using CD3/CD28 T-activator beads or antibodies. To generate iNKT cells, PBMCs are MACS-sorted via anti-iNKT microbeads labeling to enrich iNKT cells and then stimulated with αGC. To generate γδT or MAIT cells, PBMCs are stimulated with Zoledronate or 5-OP-RU, respectively. To generate NK cells, PBMCs are FACS-sorted via human CD56 antibody labeling or MACS-sorted using a human NK Cell Isolation Kit. (**C**) UCB-derived HSCs, donor bone marrow-derived HSCs, or PSCs-differentiated HSCs can be transduced with different TCRs, including tumor antigen specific TCRs (e.g., NY-ESO-1 TCR), iNKT αβ TCRs, MAIT αβ TCRs, and γδ TCRs. The gene-engineered HSCs are then put into in vitro culture systems allowing these HSCs to differentiate into mature T cells with specific TCRs. NK cells can also be differentiated from CD34^+^ HSCs using a cocktail of cytokines in vitro. The resulting T or NK cells are engineered with CARs and then expanded in vitro before infusion into patients. Abbreviations: MR1, major histocompatibility complex, class I-related protein; 5-OP-RU, 5-(2-oxopropylideneamino)-6-d-ribitylaminouracil.

**Table 1 cells-10-03497-t001:** Summary of αβ T, γδ T, iNKT, MAIT, and NK cell-based allogeneic cell products.

Immune Cell Types	Tumor Recognition Receptors	Restriction Reactivity	Staining Markers	GvHDRisk	Allogeneic Cell Products
Conventional αβ T cells	Highly diverse αβ TCRs	MHC-I and MHC-II	CD3^+^ TCR αβ^+^	High	Genome-edited, donor-derived UCART19 [25,36]
					CD19 CAR-T cells with CAR integrated into the TCR α chain [37]
					iPSC-derived CD19 CAR-T cells [38]
					In vitro generation in OP9-DL1 cultures [39]
					In vitro generation in ATO cultures [40,41]
					Rejuvenated iPSC-Derived T Cells [42,43,44,45,46,47]
Invariant natural killer T (iNKT) cells	Invariant TCR α-chain (Vα14-Jα18 in mice or Vα24-Jα18 in humans), restricted diverse TCR β-chain	CD1d	CD3^+^TCR αβ^+^6B11(iNKT TCR)^+^	Low	iPSC-derived iNKT cells [48,49]
Mucosal associated invariant T (MAIT) cells	Semi-invariant TCR α-chain (Vα19-Jα33 in mice or Vα7.2-Jα33 in humans), restricted diverse TCR β-chain	MR1	CD3^+^TCR αβ^+^Vα7.2^+^	Low	iPSC-derived MAIT cells [50,51,52]
Gamma delta (γδ) T cells	Restricted diverse γδ TCRs	Butyrophilin 3A1, CD1d	CD3^+^ TCR γδ^+^	Low	iPSC-derived γδ T cells [53]
Natural killer (NK) cells	NK activation and inhibition receptors (e.g., NKG2D, DNAM-1, KIR)	e.g., MIC-A/B, ULBP, CD155, CD112	CD3^-^CD56^+^	Low	Cord blood-derived CD19 CAR-NK cells [31,54]
					PSC-derived NK cells [55,56,57,58,59,60,61]

## Data Availability

Data sharing is not applicable to this review paper.

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
