# Peer review of "Development of Stem Cell-Derived Immune Cells for Off-the-Shelf Cancer Immunotherapies"

_cells, 2021, doi:10.3390/cells10123497_

Round 1

Reviewer 1 Report

This review attempts to do a comprehensive job of covering recent developments in derivation of novel immune cell populations from different human stem cells suitable for cancer therapy. While this review does a good job covering not only T cells but also NK cells and other more unconventional T cell populations, there are several issues that need to be clarified to make this more suitable for publication.

Main points:

  1. On line 55, the authors describe allogeneic stem cell transplants (ASCT). Of note, ASCT is more often used to abbreviate autologous stem cell transplantation. For this review, these should be more appropriately describe as allogeneic hematopoietic cell transplants or hematopoietic stem cell transplants.
  2. In line 60, the authors note that serious GVHD occurs in the majority of allogeneic transplantations. It is not clear this is currently the case as that reference dates back to 2009. With better immunosuppression protocols it is only a minority of patients that get GVHD and in most cases this is mild. This should be revised and updated with more current information.
  3. In line 63 it is suggested that there is a lack of persistence due to immunogenicity of allogeneic products. This is actually not the case in the area of allogeneic hematopoietic stem cell transplants where the donor HSCs will survive for the life of the recipient, often many years or decades. These are not immune rejected. This should be revised accordingly.
  4. On line 64, it is not clear why deficiencies in allogeneic hematopoietic cell transplant leads to favoring autologous CAR-T cell therapy. These are actually typically used for very different diseases. Autologous CAR-T cells are primarily used for B cell lymphomas where allogeneic transplant is less common. This section should be revised and corrected.
  5. In section 2 beginning on line 145, it is stated that there are 3 categories of stem cells. However, categories 2 and 3 are actually the same, as iPSCs are actually one form of pluripotent stem cells, which also include embryonic stem cells. Also please note that iPSCs does not stand for “immune cell-derived induced PSC”. Therefore, perhaps there are only 2 categories of stem cells use for allogeneic therapies.
  6. In lines 166 through 172, the authors describe derivation of NK cells using stromal cell co-culture system, though that is not actually correct for those references. References 47 and 48 use a stromal-free embryoid-body based system for initial stages and the system is completely stromal-free. There are older studies that did initially use a stromal cell-based differentiation system, for example: doi: 10.4049/jimmunol.175.8.5095 and doi: 10.1182/blood-2008-06-165225. These can be discussed if desired, though does not seen necessary. Mainly need to correct how current work is done, as in references 47 and 48.
  7. Material describing the UCART-19 cells in lines 232 and 233 repeat material that was already described earlier in lines 74 and 74. Therefore, this redundancy can be eliminated.
  8. On line 393 it is described that Fate Therapeutics developed a platform to generate NK cells from iPSCs. This work was done by the Kaufman Lab as noted in the initial publications such as references 47, 48, and 83. References 48 and 83 are redundant and references should be coordinated/corrected.
  9. In line 418 it is suggested that HSCs have limited self-renewal abilities. They should add the words in vitro after this, as HSCs have unlimited self-renewal in vivo.
  10. On Figure 2 they should also include a line on starting from pluripotent stem cells (embryonic stem cells and/or iPSCs) to produce NK cells, T cells, or other immune cells of interest. This is an important stem cell source for these allogeneic therapies and needs to be included here.

Author Response

This review attempts to do a comprehensive job of covering recent developments in derivation of novel immune cell populations from different human stem cells suitable for cancer therapy. While this review does a good job covering not only T cells but also NK cells and other more unconventional T cell populations, there are several issues that need to be clarified to make this more suitable for publication.

Main points:

  1. On line 55, the authors describe allogeneic stem cell transplants (ASCT). Of note, ASCT is more often used to abbreviate autologous stem cell transplantation. For this review, these should be more appropriately describe as allogeneic hematopoietic cell transplants or hematopoietic stem cell transplants.

Response: We appreciate the Reviewer’s comment. We have changed the abbreviation and described it as allogeneic hematopoietic stem cell transplants (Allo-HSCT).

  1. In line 60, the authors note that serious GVHD occurs in the majority of allogeneic transplantations. It is not clear this is currently the case as that reference dates back to 2009. With better immunosuppression protocols it is only a minority of patients that get GVHD and in most cases this is mild. This should be revised and updated with more current information.

Response: We appreciate the Reviewer’s constructive comment. We have revised the description:

“Within Allo-HSCT grafts, donor T cells exert potent graft-versus-tumor effects, but these same effector cells can recognize major and minor HLA complex mismatches and attack healthy host tissue. Graft-versus-host disease (GvHD) occurs in a some Allo-HSCT patients, limiting the therapeutic benefit of this treatment [13]. Although depleting these T cells from allografts reduces the GvHD risks, it leads to increased rate of tumor relapse and graft failure [14]”

  1. In line 63 it is suggested that there is a lack of persistence due to immunogenicity of allogeneic products. This is actually not the case in the area of allogeneic hematopoietic stem cell transplants where the donor HSCs will survive for the life of the recipient, often many years or decades. These are not immune rejected. This should be revised accordingly.

Response: We appreciate the Reviewer’s comment. We have revised the description:

“Host-versus-graft (HvG) responses, while not life-threatening, weaken the cell therapy before it fully executes antitumor functions [15]. Therefore, the establishment of HvG tolerance in host immune system allows durable engraftment of Allo-HSCT grafts [16]. Alemtuzumab, a humanized anti-CD52 antibody, is utilized in Allo-HSCT to alleviate severe GvHD and aid sustained hematological engraftment [17].

  1. On line 64, it is not clear why deficiencies in allogeneic hematopoietic cell transplant leads to favoring autologous CAR-T cell therapy. These are actually typically used for very different diseases. Autologous CAR-T cells are primarily used for B cell lymphomas where allogeneic transplant is less common. This section should be revised and corrected.

Response: We thank the Reviewer for pointing out. We have revised the description:

“Given the potential risks of transplant-related morbidity and mortality, Allo-HSCT is not considered as a standard therapeutic approach for blood cancers by most clinical experts and researchers, and CAR-T cell therapy gained its foothold as an autologous platform [18].”

  1. In section 2 beginning on line 145, it is stated that there are 3 categories of stem cells. However, categories 2 and 3 are actually the same, as iPSCs are actually one form of pluripotent stem cells, which also include embryonic stem cells. Also please note that iPSCs does not stand for “immune cell-derived induced PSC”. Therefore, perhaps there are only 2 categories of stem cells use for allogeneic therapies.

Response: We appreciate the Reviewer’s comment. We have corrected this section:

“Two major categories of stem cell resources could be utilized to generate allogeneic therapeutic cells: hematopoietic stem cells (HSCs) and pluripotent stem cells (PSC). HSCs are the most primitive of blood lineage cells and these cells can be obtained from umbilical cord blood (UCB), donor bone marrow, or granulocyte colony stimulating factor (G-CSF)-mobilized peripheral blood [36,37]. Established PSC lines, including embryonic stem cell (ESC) and induced pluripotent stem cell (iPSC) lines, are widely utilized for hematopoietic stem/progenitor cells and mature immune cell generation. In addition, peripheral blood mononuclear cell (PBMC)-derived immune cells including T, NK, iNKT, and MAIT cells could be reprogramed to pluripotency and then re-differentiated into functional immune cells [38–41].”

  1. In lines 166 through 172, the authors describe derivation of NK cells using stromal cell co-culture system, though that is not actually correct for those references. References 47 and 48 use a stromal-free embryoid-body based system for initial stages and the system is completely stromal-free. There are older studies that did initially use a stromal cell-based differentiation system, for example: doi: 10.4049/jimmunol.175.8.5095 and doi: 10.1182/blood-2008-06-165225. These can be discussed if desired, though does not seen necessary. Mainly need to correct how current work is done, as in references 47 and 48.

Response: We really appreciate the Reviewer’s inputs. We have corrected this section and added new references:

“To differentiate into NK cells, ESCs or iPSCs are cultured in a stromal cell-based or a stromal-free system supplemented with stem cell factor (SCF), vascular endothelial growth factor (VEGF), and bone morphogenetic protein 4 (BMP4) to induce hematopoietic differentiation, and then are switched to cultures containing IL-3, IL-15, IL-7, SCF and FLT3L to stimulate NK cell differentiation, followed by propagation with a stimulatory cell line expressing membrane-bound IL-15 (mbIL-15) or mbIL-21 [51–55].”

  1. Material describing the UCART-19 cells in lines 232 and 233 repeat material that was already described earlier in lines 74 and 74. Therefore, this redundancy can be eliminated.

Response: We thank the Reviewer for the suggestions. We have deleted the redundancy in the manuscript.

  1. On line 393 it is described that Fate Therapeutics developed a platform to generate NK cells from iPSCs. This work was done by the Kaufman Lab as noted in the initial publications such as references 47, 48, and 83. References 48 and 83 are redundant and references should be coordinated/corrected.

Response: We thank the Reviewer for pointing out the mistakes. We have corrected the references.

  1. In line 418 it is suggested that HSCs have limited self-renewal abilities. They should add the words in vitro after this, as HSCs have unlimited self-renewal in vivo.

Response: We appreciate the Reviewer’s comment. We have corrected the sentence.

  1. On Figure 2 they should also include a line on starting from pluripotent stem cells (embryonic stem cells and/or iPSCs) to produce NK cells, T cells, or other immune cells of interest. This is an important stem cell source for these allogeneic therapies and needs to be included here.

Response: We appreciate the Reviewer’s constructive comment. We have modified Figure 2 and added PSC part.

Reviewer 2 Report

Development of Stem Cell-Engineered Off-The-Shelf Cell Therapy for Cancer

This review examines the development of stem cell-engineered allogeneic cell therapies that aim to overcome the limitations of current autologous and allogeneic cell therapies, with a special focus on stem cell-engineered conventional αβ T cells, unconventional T (iNKT, MAIT, and γδ T) cells, and natural killer (NK) cells.

In general, I believe that the review will be useful and in demand by a large number of researchers engaged not only in the field of cancer biology/immunology, but also by scientists conducting research in the field of cell therapy. Overall, the manuscript is well written as a systemic review. This manuscript gives a detailed analysis of this area of research, listing the main achievements, breakthrough results and prospects. However, please add a brief table with some relevant information to compare the cell-based therapy using different immune cells and their subtypes. Including methods and mechanisms will provide more enriched and interesting data for global readers.

Author Response

Response: We appreciate the Reviewer’s constructive comment. We have incorporated a table in the manuscript and compared different cell-based therapy (See Revised Manuscript Table 1).

Reviewer 3 Report

The manuscript “Development of Stem Cell-Engineered Off-The-Shelf Cell Therapy for Cancer” is focused on Cell-based cancer immunotherapy like CAR-T cell therapies used for the treatment of hematological malignancies. This manuscript has its own merits but as described by authors that Allogeneic cell products have GvHD concerns should be elaborated more by providing more details of donor-specific antibodies generation by using these products. Besides, the current form of the manuscript needs some clarifications before publication.

  1. Out of the 3 proposed stem cell resources for allogeneic therapy, there is no mention of MSCs. Considering their immune evasive and migratory property, they are being used in targeted therapies against cancer (doi: 3389/fbioe.2020.00043, doi.org/10.1186/s12935-021-01836-9)
  2. The title does not do justice to the text of the manuscript since the review only focuses on stem cell-derived allogeneic therapies in the context of immunotherapies alone. There are other types of therapies as well (https://www.nature.com/articles/s41417-020-0179-6, https://www.science.org/doi/10.1126/sciadv.abe8671). So, either change the title to focus more on the immunotherapies or include more data on different stem cell-derived therapies as described in the recent study https://doi.org/10.1016/j.lfs.2021.119465 .
  3. More information needs to be added regarding the off-shelf products as described in the introduction. Also, their current status, ongoing/completed clinical trials, and whether any of them are being sold or used commercially.
  4. Engineering stem cells for allogeneic therapy should elaborate more on the engineered stem cells currently being used and the associated clinical trials, rather than only focusing on stem cell-derived immune cells.
  5. Page 2, Line 68: Justify why to focus on HLA-E rather than conventional other HLA subtypes like A, B, C, or Class II.
  6. The information given in the review does not seem novel. There are some recently reported reviews with the same information -https://www.frontiersin.org/articles/10.3389/fimmu.2020.583716/full, https://www.frontiersin.org/articles/10.3389/fimmu.2019.02250/full

Author Response

  1. Out of the 3 proposed stem cell resources for allogeneic therapy, there is no mention of MSCs. Considering their immune evasive and migratory property, they are being used in targeted therapies against cancer (doi: 3389/fbioe.2020.00043, doi.org/10.1186/s12935-021-01836-9)

We appreciate the reviewer’s comment. Given our expertise and the scope of this review, we have not covered MSCs but we have included a brief description and the two sources highlighting their potential in the discussion. We have altered our title to reflect this.

  1.  The title does not do justice to the text of the manuscript since the review only focuses on stem cell-derived allogeneic therapies in the context of immunotherapies alone. There are other types of therapies as well (https://www.nature.com/articles/s41417-020-0179-6, https://www.science.org/doi/10.1126/sciadv.abe8671). So, either change the title to focus more on the immunotherapies or include more data on different stem cell-derived therapies as described in the recent study https://doi.org/10.1016/j.lfs.2021.119465 .

We appreciate the reviewer’s comment. We have altered our title to: Development of Stem Cell-Derived Immune Cells for Off-The-Shelf Cancer Immunotherapies

  1.  More information needs to be added regarding the off-shelf products as described in the introduction. Also, their current status, ongoing/completed clinical trials, and whether any of them are being sold or used commercially.

We appreciate the reviewer’s comment. We have referenced lead allogeneic T, NK, iNKT, and gamma delta T cell candidates derived from fully differentiated cells and their clinical trials. We have now clarified that these treatment modalities are in early-phase clinical studies.

  1.  Engineering stem cells for allogeneic therapy should elaborate more on the engineered stem cells currently being used and the associated clinical trials, rather than only focusing on stem cell-derived immune cells.

We appreciate the reviewer’s comment. We have changed our title to reflect our focus on stem cell-derived immune cells and have included ongoing clinical trials (such as Fate Therapeutics trials). 

  1.  Page 2, Line 68: Justify why to focus on HLA-E rather than conventional other HLA subtypes like A, B, C, or Class II.

We appreciate the reviewer’s comment. In the previous sentence, we refer to CRISPR KO of HLA Class I and II molecules to alleviate T cell rejection. We subsequently mention the introduction HLA-E and other NK cell injection receptors to prevent NK cell rejection. We include information to address both host T and NK cell-mediated rejection of the allogeneic cell therapy.

  1.  The information given in the review does not seem novel. There are some recently reported reviews with the same information -https://www.frontiersin.org/articles/10.3389/fimmu.2020.583716/full, https://www.frontiersin.org/articles/10.3389/fimmu.2019.02250/full

We appreciate the reviewer’s comment. We feel our review is different from the two reviews referenced in this comment, as the two papers do not discuss stem cell-derived immune cell therapies. These two significant reviews have been cited in our manuscript.

Round 2

Reviewer 1 Report

There remain several items in the previous review that have not been appropriately addressed, as outlined below. The authors should get another co-author or consultant with expertise in human clinical BMT (HSCT) to review and appropriately revise the portions that have to do with GVHD, clinical BMT/HSCT, etc. Use of references a decade or more out-of-date do not lead to a suitable review on these topics.

Specific points:

  1. In line 60, the authors note that serious GVHD occurs in the majority of allogeneic transplantations. It is not clear this is currently the case as that reference dates back to 2009. With better immunosuppression protocols it is only a minority o fpatients that get GVHD and in most cases this is mild. This should be revised and updated with more current information.

Response: We appreciate the Reviewer’s constructive comment. We have revised the description:

“Within Allo-HSCT grafts, donor T cells exert potent graft-versus-tumor effects, but these same effector cells can recognize major and minor HLA complex mismatches and attack healthy host tissue. Graft-versus-host disease (GvHD) occurs in a some Allo-HSCT patients, limiting the therapeutic benefit of this treatment[13]. Although depleting these T cells from allografts reduces the GvHD risks, it leads to increased rate of tumor relapse and graft failure [14]”

- REPLY: This is very narrow comment on possible treatment or prevention of GVHD. T cell depletion is not specifically done in clinical care. More typical is use of standard immunosuppressove mediaction such as calcineurin inhibitors (Cyclosporin or tacrolimus) and/or methotrexate treatment and/or post-transplant cyclophosphamide, etc. While the risk of GVHD must be balanced with the benefit of graft-versus-tumor effect, this is manageable in most cases. Again, a more recent reference than 2009 and more accurate description of the manageable risks of GVHD in allo-HSCT is needed here.

  1. In line 63 it is suggested that there is a lack of persistence due to immunogenicity of allogeneic products. This is actually not the case in the area of allogeneic hematopoietic stem cell transplants where the donor HSCs will survive for the life of the recipient, often many years or decades. These are not immune rejected. This should be revised accordingly.

Response: We appreciate the Reviewer’s comment. We have revised the description:

“Host-versus-graft (HvG) responses, while not life-threatening, weaken the cell therapy before it fully executes antitumor functions[15]. Therefore, the establishment of HvG tolerance in host immune system allows durable engraftment of Allo-HSCT grafts[16]. Alemtuzumab, a humanized anti-CD52 antibody, is utilized in Allo-HSCT to alleviate severe GvHD and aid sustained hematological engraftment [17].

REPLY: Again, this reply is not accurate for typical clinical care. Alemtuzumab (anti-CD52) is not a common (and not approved) treatment to prevent GVHD. Again, this reference #17 is from 2005. More updated reference(s) are needed here and the authors should not HvG somehow damages the transplanted cells. In reality, 1000s of patients per year have successful allo-HSCT, so more accurate info here is needed.

  1. On line 64, it is not clear why deficiencies in allogeneic hematopoietic cell transplant leads to favoring autologous CAR-T cell therapy. These are actually typically used for very different diseases. Autologous CAR-T cells are primarily usedfor B cell lymphomas where allogeneic transplant is less common. This section should be revised and corrected.

Response: We thank the Reviewer for pointing out. We have revised the description:

“Given the potential risks of transplant-related morbidity and mortality, Allo-HSCT is not considered as a standard therapeutic approach for blood cancers by most clinical experts and researchers, and CAR-T cell therapy gained its foothold as an autologous platform [18].”

REPLY: This response is just wrong. Allo-HSCT is indeed standard of care for relapsed hematologic malignancies such as leukemia (AML and ALL) or certain lymphomas, either in CR1 or after relapses depending on the disease, patient age, comorbidities, etc. This needs to be revised appropriately.

  1. On line 393 it is described that Fate Therapeutics developed a platform to generate NK cells from iPSCs. This work was done by the Kaufman Lab as noted in the initial publications such as references 47, 48, and 83. References 48 and 83 are redundant and references should be coordinated/corrected.

Response: We thank the Reviewer for pointing out the mistakes. We have corrected the references.

REPLY: This was not corrected in the text. Line 444 still states that Fate Therapeutics developed iPSC-NK cells. In reality, it was the Kaufman group that was the first to do this and this should be recognized as such. First this was done with hESCs (refs. 57 and 58) and then from human iPSCs (refs 54 and 55). Fate Therapeutics used this know-how to bring iPSC-NK cells to clinical trials. Also, references 58 and 82 are still repeated.
